# Advances in the Asymmetric Synthesis of BINOL Derivatives

**DOI:** 10.3390/molecules28010012

**Published:** 2022-12-20

**Authors:** Everton Machado da Silva, Hérika Danielle Almeida Vidal, Marcelo Augusto Pereira Januário, Arlene Gonçalves Corrêa

**Affiliations:** Centre of Excellence for Research on Sustainable Chemistry, Department of Chemistry, Federal University of São Carlos, São Carlos 13565-905, Brazil

**Keywords:** BINOL, chirality transfer, metal-mediated enantioselective coupling, organocatalysis, kinetic resolution

## Abstract

BINOL derivatives have shown relevant biological activities and are important chiral ligands and catalysts. Due to these properties, their asymmetric synthesis has attracted the interest of the scientific community. In this work, we present an overview of the most efficient methods to obtain chiral BINOLs, highlighting the use of metal complexes and organocatalysts as well as kinetic resolution. Further derivatizations of BINOLs are also discussed.

## 1. Introduction

The chirality resulting from restricted rotation around a single bond is called atropisomerism (axial chirality). This phenomenon was first described by Christie and Kenner [1] in 1922 when investigating the biaryl 6,6’-dinitro-2,2’-diphenic acid (Figure 1), and the term “atropisomer”, derived from the Greek where “a” means “not” and “tropes” means “turn”, was created by Kuhn. Atropisomers belong to the class of axially chiral compounds; however, in this case, the enantiomers exist due to restricted rotation around a single bond [2].

Axial chirality has also been considered as an important structural element of many natural products [3] and bioactive compounds, whose enantiomers generally exhibit different pharmacological activities and metabolic processes in vivo and in vitro [4]. Examples of natural products include viriditoxin, produced by fungi with antibacterial activity, and vancomycin, an amphoteric glycopeptide antibiotic produced by soil bacterium (Figure 2).

Due to their relevant biological properties, the asymmetric synthesis of atropisomers has attracted the interest of the scientific community. Furthermore, atropisomers have been important chiral ligands since the 1980s, when BINAP was developed for enantioselective reactions catalyzed by transition metals [3].

Although conventional chiral resolution of racemates and chiral auxiliary reactions are generally used for the construction of axially chiral compounds in enantiomerically pure form, asymmetric catalysis meets the demand for high efficiency and economic value [5]. The use of atropisomeric compounds as ligands in metal-mediated catalysis has revolutionized the organometallic chemistry and asymmetric synthesis fields. Due to the high demand and importance of these chiral biaryl scaffolds, several synthetic procedures [6], reviews [7,8,9], and concepts [10] have been published in the literature.

Axially chiral biaryl auxiliaries and catalysts (such as BINOL or BINAP) exhibit excellent chirality transfer properties [11]. Due to the importance of axially chiral biaryl compounds, several interesting methods for their directed atroposelective construction have been developed. Table 1 shows some catalysts used for the transfer of axial chirality, bearing in mind that the substrates used do not have any type of chirality, thus strengthening the excellent transfer of this property.

Herein, we present an overview of the most efficient methods for asymmetric synthesis of chiral BINOLs reported in the last two decades, highlighting the use of metal complexes and organocatalysts, as well as kinetic resolution and further derivatizations.

## 2. Synthesis of BINOLs Skeleton

### 2.1. Metal-Mediated Oxidative Enantioselective Coupling

Among the available methods for the synthesis of optically active BINOLs, one of the most explored is the oxidative dimerization of 2-naphthols mediated by complexes of Cu [39,40,41,42,43,44,45,46] Fe [47,48,49], V [50,51,52,53,54,55], Ru [56], and chiral ligands (very often amines), normally generated in situ. In this regard, excellent reviews discussing these methods have been reported by Brunel [57], Wang [58], Bryliakov et al. [59], and Liao et al. [60] (Figure 1 and Figure 2).

Oxidative coupling may occur through three different mechanisms: (1) radical–radical coupling, (2) heterolytic coupling of cationic species with 2-naphthol, or (3) radical–anion coupling, the latter generally being the most accepted to support this type of transformation [55,61,62,63,64]. An important step is the one where the catalytic species is complexed with directing groups or coordination assistants at the C3 position of 2-naphthols—notably ester groups [41,45]—which, in many cases, is a “sine qua no” condition for the success of the synthetic protocols (Figure 3).

With due recognition of the particularities of each case, the radial–anion mechanism [55,61,62,63,64,65] (Figure 3) usually proceeds via generation of the radical species **B** resulting from an oxidation of 2-naphthol **A** by a metal catalyst (M^n+^). **B** is then added to another neutral 2-naphthol molecule to form a new C-C bond and generate the **C**-radical, which is further oxidized by O_2_ to restore aromaticity.

The most recent methods for obtaining enantiomerically pure BINOLs are still based on the catalytic dyad metal–chiral ligands. In this sense, Chen and colleagues [66] (Figure 4) have developed a new chiral 1,5-*N*,*N*-bidentate ligand based on a spirocyclic skeleton of pyrrolidine oxazoline and CuBr to couple 2-naphtols **3b**. The efficient catalytic species formed in situ allows for (*S*)-BINOL derivatives (**1**) with high enantioselectivity (up to 99% ee) and good yields (up to 87%) to be obtained. Based on experimental results and the literature, the authors proposed that this coupling proceeds via radical–anion coupling, where the complex generated in situ coordinates to form species **D** in the presence of air, which couples with radical **E** (generated through an electron transfer from the outer sphere with another Cu(II) complex) to form the intermediate **F** (Figure 5). The coupling product is obtained after tautomerization of **H**. The authors found experimental evidence that, during the coupling process, the attack from **E** to **F** by the *Si* face was favored, probably because of the greater steric impediment to attack by the *Re* face.

Che and co-workers introduced a chiral aminopyridine-like ligand—bisquinolyldiamine [(1*R*,2*R*)-*N*^1^,*N*^2^-di(quinolin-8-yl)cyclohexane-1,2-diamine (BQCN)]— and applied it to the iron-catalyzed asymmetric *cis*-dihydroxylation of alkenes [67]. Inspired by this work, Liu’s group [68] (Figure 6) established a methodology for the asymmetric oxidative homo-coupling of 2-naphthols (**3c**), leading to the synthesis of (*S*)-BINOL derivatives (**1**) mediated by a Fe complex and generated in situ from Fe(ClO_4_)_2_ and the BQCN ligand. Excellent yields (up to 99%) and enantiomeric excesses (up to 81%) have been reported.

From the same perspective, Uchida’s group [69] (Figure 7) developed remarkable enantioselective aerobic coupling between 2-naphthols **3d** in the presence of the (aqua)ruthenium complex (salen). The protocol provided (*R*)-BINOLs (**1**) with yields between 55 and 85% and enantiomeric excesses up to 94%. Through mechanistic studies, these researchers concluded that, in this case, cross-coupling selectivity is dominated by steric rather than electronic effects, which can be controlled by chemoselective oxidation via single electron transfer (SET) and oxidative carbon–carbon bond formation, a process for which ruthenium(salen) catalyst proved to be suitable [62]. Therefore, the authors have proposed that this transformation proceeds via oxidation of one of the coupling partners to the electrophilic intermediate radical **I**, which is converted to the desired BINOL after chemoselective coupling [62].

Recently, Subramanian et al. [70] (Figure 8) developed a Cu(II)-2+4-μ4-oxo tetranuclear open frame macrocyclic/BINAN complex and employed it in the asymmetric oxidative coupling of 2-naphthols **3e**, obtaining (*R*)-BINOL derivatives (**1**) with good to excellent yields (70–96%) and enantiomeric excesses between 68 and 74%.

Ishihara and co-workers [71] (Figure 9) developed a method for enantioselective oxidative coupling of 2-naphthol derivatives **3d** in the presence of a chiral Fe(II)-diphosphine oxide complex. The products of interest were obtained with yields up to 98 % and enantiomeric excesses between 60 and 85%.

A copper catalyst prepared in situ from a ligand synthesized by the fusion of chelating picolinic acid/substituted BINOLs and CuI was employed in the asymmetric oxidative coupling of 2-naphthols (**3e**). In this work, published by Zhang et al. [72] (Figure 10), 6,6’-disubstituted (*R*)-BINOLs (**1**) were obtained with yields of up to 89% and excellent enantioselectivities (up to 96% ee). The reaction was accompanied by Mass Spectroscopy, and identification of a peak corresponding to the complex **J** allowed the authors to propose a mechanism pathway through the transition state **K**.

Continuing work involving multifunctional chiral catalysis via double activation, Takizawa’s group [73] developed complexes **A**-**C** (Figure 11)—from VOSO_4_ and Schiff base ligands generated via condensation of (*S*)-tert-leucine and 3,3 ‘-formyl-(R)-BINOL—which have been successfully applied in the synthesis of (*R*)- and (*S*)-BINOL (**1**) with yields between 46 and 76%, in addition to enantiomeric excesses of up to 91%.

### 2.2. Electrochemical Synthesis

Despite the inherent advantages of electrochemical synthesis, notably in terms of sustainability [74], few examples of enantioselective coupling for the construction of chiral BINOLs have been reported so far. In 1994, which appears to be the first record of this type of synthesis, Bobbitt et al. [75] (Figure 12) established a method for enantioselective coupling of 2-naphthols (**3f**) on a TEMPO-modified graphite electrode in the presence of (-)-sparteine in acetonitrile to afford (*S*)-BINOL (**1**) with excellent yields and enantiomeric excesses.

Recently, Mei’s group [74] (Figure 13) demonstrated the first example of a Ni-catalyzed enantioselective electrochemical reductive coupling of 2-naphtols (**4**) in an undivided cell for the construction of axially chiral BINOL derivatives (**1**) with good yields (up 91%) and enantiomeric excess of up to 98%.

### 2.3. Organocatalyzed Synthesis/Kinetic Resolution of BINOLs/BINAPs

Among the methods for the synthesis/kinetic resolution (KR) of axially chiral biaryl and binaphthyl compounds, the organocatalyzed approach has been more explored in the last decade. Some examples were described in the review by Cheng et al. [76]. In this topic, the kinetic synthesis/resolution of BINOL skeletons will be addressed using organocatalysts that do not have axial chirality, thus providing induction to the products of interest.

In 2005, Tsuji and co-workers [77] reported a kinetic resolution of 2,2’-dihydroxy-1,1’-biaryls using a palladium-catalyzed atroposelective alcoholysis of racemic vinyl ethers (**5**). The method uses the organocatalyst (*R*,*R*)-1,2-cyclohexanediamine (**6**) and a mixture of methanol and dichloromethane as solvent at 20 °C. Five examples were obtained and it was observed that the volume of the acyl group directly influences selectivity, as shown in Figure 14 where the larger substituent (1-adamantyl) provided high selectivity.

In 2012, Dan and co-workers [78] evaluated the Ferrier-type rearrangement using chiral bicyclic guanidine as a catalyst; however, the reaction was sluggish, affording an optically active product with low yield and enantioselectivity (Figure 15A). Upon these results, the authors used another strategy, based on studies by Masatoshi and Reiko [79] for synthesis of the optically active biaryl through optical resolution of the corresponding racemate using chiral diamine (**12**) (Figure 15B). Initially, there was deprotection of the methyl ether, and in the second step the racemic binaphthol derivative was recrystallized from toluene in the presence of (*S*,*S*)-1,2-diphenyl-1,2-ethanediamine (**12**), leading to compound (*R*)-**11** with 95% ee and 22% yield.

In 2014, Sibi and co-workers [80] proposed a new kinetic resolution that employs a chiral 4-(dimethylamino)pyridine (DMAP) derivative **14** as a catalyst via *O*-acylation (Figure 16). The method proved to be efficient, and both secondary alcohols and axially chiral biaryl compounds were obtained with selectivity factors of up to 37 and 51, respectively. Increased conversion was also observed for binaphthyl substrates with an electron-rich group in the *ortho* position.

In 2014, Zhao and co-workers [81] reported an atroposelective kinetic resolution using *N*-heterocyclic carbene (NHC) **18** as a catalyst to resolve the enantiomers of BINOL (Figure 17). The 1,1’-biaryl-2,2’-diol derivatives **16** were explored, where the products obtained, both acylated **19** and the recovered BINOL **16**, had high selectivity.

Sparr and Link [82] described a highly enantioselective synthesis of binaphthyl **20** by intramolecular aldol condensation using (*S*)-pyrrolidinyl-tetrazole **21** as a catalyst (Figure 18). The authors described that the high selectivity of the process stems from the efficient transfer of stereochemical information from the catalyst into the axis of chirality of biaryl products. The examples obtained showed good yields and high enantiomeric ratios.

In 2017, Shirakawa and co-workers [83] reported a highly enantioselective organocatalytic method for the synthesis of atropisomeric biaryls through cation-directed *O*-alkylation. Reaction of racemic 1-aryl-2-tetralones (**23**) with the ammonium salt **24** (obtained from chiral quinidine) under basic conditions and using an alkylation agent leads to highly enantioselective *O*-alkylation (up to 98:2 er). According to the proposed mechanism, the basic medium initially promotes deprotonation and makes it possible to generate two enolate enantiomers associated with the chiral salt that form diastereomeric ion pairs; however, the chiral ammonium counterion is capable of rapidly differentiating balancer atropisomeric enolates, leading to highly atropselective *O*-alkylation. The in situ oxidation step with 2,3-Dichloro-5,6-dicyano-1,4-benzoquinone (DDQ) then takes place to obtain the BINOL derivatives (**25**) without loss of the enantiomeric ratio. The authors also noted that enantioselectivity can be controlled by the catalyst structure and the type of base and solvent used. Under optimized conditions, they obtained a broad scope of 20 examples with moderate to excellent yields (Figure 19).

In 2019, Sparr and co-workers [84] described a non-canonical polyketide cyclization, affording atropisomeric tetra-*ortho*-substituted binaphthalenes. Using hexacarbonyl substrates **26** and aminoethanol-derived proline-based catalyst **27** via the cascade of two arene ring formation reactions (Figure 20), it was possible to obtain enantioenriched tetra-*ortho*-substituted binaphthalenes **28** through atroposelective aldol condensation. The explanation of the mechanism was based on NMR studies, where it was observed that two sequential aldol additions take place before a double dehydration, hence alleviating the acute nonbonding interaction during C−C bond formation. With product **28** (without substituents), it was possible to perform a derivatization through a triflation and arene coupling, obtaining (*S*)-**29** with 84% yield (over two steps) and >99:1 er.

Based on a previous work, where high atroposelectivity was acquired starting from racemic 2-tetralones, Jones et al. [85] extended the strategy for the use of BINOLs (Figure 21). The reaction proceeds under basic conditions using cinchona derived catalyst **32** in a mixture of toluene and ethyl ether at room temperature for 48 h, leading to the formation of compound (*R*)-**30** with 47% yield and 96% ee and compound (*S*)-**31** with 49% yield and 80% ee.

The reversible deprotonation of compound **30** leads to the formation of a diastereoisomeric BINOlate ammonium salt, which reacts at different rates in the alkylation step. The proposed transition state involves a hydrogen bonding of the ammonium salt with the secondary alcohol of the BINOlate anion, and an additional hydrogen bonding of the benzyl electrophile to the methyl ether.

### 2.4. Enzymatic Kinetic Resolution of BINOL

For decades, enzymatic kinetic resolution was considered the most reliable strategy for obtaining optically enriched compounds. In 1989, the first efficient method for enzymatic resolution of rac-BINOL (**33**) was described by Kazlaukas [86], which was based on enantiospecific hydrolysis catalyzed by cholesterol esterase (Figure 22). The procedure uses low-cost bovine pancreatic acetone powder (PAP). The enantiomer (*S*)-**1** was obtained with 66% yield and 99% ee and recrystallized with toluene, with compound (*R*)-**33** formed after hydrolysis; enantiomer (*R*)-**1** was obtained with 63% yield and 99% ee after filtration and being washed with cold toluene.

The potential of enantioselective kinetic resolution to prepare atropisomeric compounds was initially proven by designing an enzymatic kinetic resolution, as illustrated by the elegant work of Aoyagi et al. [87]. Lipase (*Candida antarctica*) was used to catalyze the hydrolysis of BINOL monoester-derivatives **34**, affording (*R*)-**1** and (*S*)-**34** with excellent yields and good enantioselective excesses (Figure 23).

In 2018, Moustafa and co-workers [88] described the lipase-catalyzed KR process, which uses immobilized *Pseudomonas* sp. lipoprotein lipase (Toyobo LIP301). Lipase selectively catalyzes readily available racemic substrates and provides stability to acylated products against racemization, promoting the formation of product (*R*)-**35** with 24% yield and 99% enantiomeric excess (Figure 24).

### 2.5. Chemical Derivatizations on the BINOL Skeleton

As previously mentioned, most chiral catalysts with the BINOL backbone are now commercially available. In this section, we will discuss the synthesis of new chiral catalysts.

Since 1999, Maruoka’s group [89,90] (Figure 25) has synthesized a series of chiral phase transfer catalysts based on quaternary ammonium bromides salts prepared from 1-(*S*)-BINOL, which have been successfully applied in the synthesis of natural and unnatural amino acids. In the same context, the group reported, in 2013 [12], the preparation of catalyst **40** (via insertion of piperazine **37**) from the brominated strategic derivative **36**, with 49% yield. In this case, the new chiral auxiliary was employed in the asymmetric phase transfer functionalization of 1-alkylalene-1,3-dicarboxylates with *N*-arylsulfonyl imines and allylic/benzyl bromides for the preparation of tetrasubstituted allenes.

Recently, Schaus’ group [91] designed and prepared the (*S*)-3,3’,6,6’-tetrakis(trifluoromethyl)-BINOL **42** from *(S*)-**1** by the subsequent protection, bromination, and insertion of the CF_3_ groups into the desired position (Figure 26). This chiral catalyst was employed in the asymmetric synthesis of 1,3-substituted chiral allenes via boronate addition to sulfonyl hydrazones.

For the catalysts based on BINOL derivatives with phosphoric acid, the most efficient method described in the literature for the insertion of phosphoric acid is phosphorylation with POCl_3_ (Figure 27), which can be performed in both enantiomerically pure and racemic BINOL [92,93].

## 3. Conclusions

There have been significant advances in the aerobic enantioselective coupling methods of 2-naphthols for the synthesis/kinetic resolution of chiral BINOLs via transition metal-mediated, electrochemical, organocatalytic, and enzymatic resolution. However, it was evident that there are still challenges to be overcome in both fields, such as the relatively high reaction time in some cases and the use of toxic reagents and solvents such as dichloromethane and toluene. In addition, as already highlighted by other authors, the mechanistic discussions that govern asymmetric induction in these cases are still preambular, denoting the need for deepened theoretical exploration in this particular area.

## Data Availability

Not applicable.

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
