# Peer review of "Advances in the Asymmetric Synthesis of BINOL Derivatives"

_molecules, 2022, doi:10.3390/molecules28010012_

Round 1

Reviewer 1 Report

The authors review the asymmetric synthesis of BINOL derivatives. It is a timely and appropriate review which covers most of the asymmetric synthesis reported on BINOLs. The manuscript is recommended for publication after the authors address the following concerns/comments.

1) The following reference may be appropriately cited and the contents briefly discussed.

Acc. Chem. Res. 2022, 55, 20, 2949–2965

2) The drawings in the manuscript are really terrible, the size of the benzene ring is inconsistent, the size of the bond is not aesthetically pleasing, and the units of measurement are also faulty, such as sometimes using “yield” and sometimes using “yielded” or “yields”. And “10% mol” or “10 mol%”. The authors should carefully check the diagrams to ensure the correct representation, which is also a kind of respect to the academic.

3) Table 1, ref.[30-33] and [38], ‘’2,4,6-(iPr)3C6H2’’ should be replaced by ‘’2,4,6-(i-Pr)3C6H2’’

4) Scheme 4, 6, 7 and 16, “MS 4A” should be replaced with “MS 4Å”, and “24 °c” should be replaced with “24 °C”

5) A space should normally be provided between figures and units in academic papers. Such as “-78°C” should be replaced with “-78 °C”.

6) P13, DDQ needs to be expressed in its full name: 2,3-Dichloro-5,6-dicyano-1,4-benzoquinone (DDQ).

7) Schemes 6 and 12: The ligand number should be bolded.  L.5: R= Et

8)Ref. 50, “Euro. J. Org. Chem.” should be replaced by “Eur. J. Org. Chem.”

9) There are many other minor errors in the manuscript, so please check it again by all means. Especially Scheme 1 and Scheme 2.

Author Response

The authors review the asymmetric synthesis of BINOL derivatives. It is a timely and appropriate review which covers most of the asymmetric synthesis reported on BINOLs. The manuscript is recommended for publication after the authors address the following concerns/comments.

Thank you.

1) The following reference may be appropriately cited and the contents briefly discussed.

Acc. Chem. Res. 2022, 55, 20, 2949–2965

We add the reference as ref. 81.

2) The drawings in the manuscript are really terrible, the size of the benzene ring is inconsistent, the size of the bond is not aesthetically pleasing, and the units of measurement are also faulty, such as sometimes using “yield” and sometimes using “yielded” or “yields”. And “10% mol” or “10 mol%”. The authors should carefully check the diagrams to ensure the correct representation, which is also a kind of respect to the academic.

The formatting error was corrected as requested.

3) Table 1, ref.[30-33] and [38], ‘’2,4,6-(iPr)3C6H2’’ should be replaced by ‘’2,4,6-(i-Pr)3C6H2’’

The formatting error was corrected as requested.

4) Scheme 4, 6, 7 and 16, “MS 4A” should be replaced with “MS 4Å”, and “24 °c” should be replaced with “24 °C”

The formatting error was corrected as requested.

5) A space should normally be provided between figures and units in academic papers. Such as “-78°C” should be replaced with “-78 °C”.

The formatting error was corrected as requested.

6) P13, DDQ needs to be expressed in its full name: 2,3-Dichloro-5,6-dicyano-1,4-benzoquinone (DDQ).

The full name of the compound was added.

7) Schemes 6 and 12: The ligand number should be bolded.  L.5: R= Et

The formatting error was corrected as requested.

8)Ref. 50, “Euro. J. Org. Chem.” should be replaced by “Eur. J. Org. Chem.”

The formatting error was corrected as requested.

Reviewer 2 Report

                 The review describes “Advances on the Asymmetric Synthesis of BINOL Derivatives”. The review covered various methods of synthesis of BINOL derivatives such as coupling, kinetic resolution, electrochemistry, organocatalysis etc. Considering the importance and application of BINOL derivatives as well as readers interest, this review can be accepted for publication in this journal with minor corrections.

 Corrections:

1) The first three lines of abstract is too broad, feels like introduction rather than abstract. 

2) Authors can provide structure of biaryl 6,6’-dinitro-2,2’-diphenic acid while describing the definition in the introduction for the benefit of young readers. 

3) Figure titles reads as “commercially available drugs”, can authors provide a reference to prove that viriditoxin is a commercial drug, as the statement looks misleading. 

4) In the introduction the authors should clearly mention a timeline from which year publication, this review focuses as the title is “Advances on the”.

5) In table 1, there is no need for use of term Z in the R2/R3 column, rather they can give structure.

6) In Scheme 1, for equation I, II and III structure of chiral amines can be given.

7) Throughout the review in the schemes R, R1, R2 etc should be defined.

8) In Scheme 1, equation X is not there. To avoid confusion, it must be corrected.

9) In Scheme 2, in reaction condition XII and XIX subscript errors to be rectified.

10) In Scheme 8, structure of copper complex to be checked and define X in the structure.

11) In Scheme 9, oxygen is not connected to any atom.

12) Why there is section 2.1.2 after 2.1 rather than 2.2?

13) In scheme 18, check for absence of enol double bond before DDQ step.

14) In Scheme 21, more details such as reaction time and other conditions can be mentioned.

15) In Scheme 24, Pd(OAc)2 is misspelled.

16) In conclusion the authors can avoid giving reference number. They can modify the sentence if needed.

Author Response

The review describes “Advances on the Asymmetric Synthesis of BINOL Derivatives”. The review covered various methods of synthesis of BINOL derivatives such as coupling, kinetic resolution, electrochemistry, organocatalysis etc. Considering the importance and application of BINOL derivatives as well as readers interest, this review can be accepted for publication in this journal with minor corrections.

Thank you.

Corrections:

1) The first three lines of abstract is too broad, feels like introduction rather than abstract.

The abstract it was reformulated.

2) Authors can provide structure of biaryl 6,6’-dinitro-2,2’-diphenic acid while describing the definition in the introduction for the benefit of young readers.

The structure of the compound was added.

3) Figure titles reads as “commercially available drugs”, can authors provide a reference to prove that viriditoxin is a commercial drug, as the statement looks misleading.

In this case the tittle has an error, but it was corrected.

4) In the introduction the authors should clearly mention a timeline from which year publication, this review focuses as the title is “Advances on the”.

We have included in the reviewed manuscript: “Herein, we present an overview of the most efficient methods for asymmetric synthesis of chiral BINOLs, highlighting the use of metal complexes and organocatalysts, as well as kinetic resolution and further derivatizations, reported in the last two decades.”

5) In table 1, there is no need for use of term Z in the R2/R3 column, rather they can give structure.

The formatting error was corrected as requested.

6) In Scheme 1, for equation I, II and III structure of chiral amines can be given.

The structure of these compounds was added.

7) Throughout the review in the schemes R, R1, R2 etc should be defined.

The formatting error was corrected as requested.

8) In Scheme 1, equation X is not there. To avoid confusion, it must be corrected.

The X was added.

9) In Scheme 2, in reaction condition XII and XIX subscript errors to be rectified.

The formatting error was corrected as requested.

10) In Scheme 8, structure of copper complex to be checked and define X in the structure.

The formatting error was corrected as requested. The X was added.

11) In Scheme 9, oxygen is not connected to any atom.

The formatting error was corrected as requested.

12) Why there is section 2.1.2 after 2.1 rather than 2.2?

The formatting error was corrected as requested.

13) In scheme 18, check for absence of enol double bond before DDQ step.

The formatting error was corrected as requested.

14) In Scheme 21, more details such as reaction time and other conditions can be mentioned.

More details were added.

15) In Scheme 24, Pd(OAc)2 is misspelled.

The formatting error was corrected as requested.

16) In conclusion the authors can avoid giving reference number. They can modify the sentence if needed.

The formatting error was corrected as requested.

Reviewer 3 Report

This manuscript summarized the progression of methods for the asymmetric synthesis of chiral BINOLs, highlighted the use of metal complexes and organocatalysts, as well as kinetic resolution and further derivatizations. It's a valuable review for the future development of BINOL derivatives and methods, however, several issues should be addressed before the acceptance of publication.

1)Title for table 1 is incorrect.

2)In Scheme 9 and 10, correct the L.4 to L4, and L.5 to L5

3)In scheme 24, there are extra positive charge marks in the structure for 38 and 39.

4)At the end of the references part, there is reference 102 which should be deleted.

Author Response

This manuscript summarized the progression of methods for the asymmetric synthesis of chiral BINOLs, highlighted the use of metal complexes and organocatalysts, as well as kinetic resolution and further derivatizations. It's a valuable review for the future development of BINOL derivatives and methods, however, several issues should be addressed before the acceptance of publication.

Thank you.

1)Title for table 1 is incorrect.

The formatting error was corrected as requested.

2)In Scheme 9 and 10, correct the L.4 to L4, and L.5 to L5

The formatting error was corrected as requested.

3)In scheme 24, there are extra positive charge marks in the structure for 38 and 39.

The formatting error was corrected as requested.

4)At the end of the references part, there is reference 102 which should be deleted.

The formatting error was corrected.